# Process Evaluation of the ‘No Money No Time’ Healthy Eating Website Promoted Using Social Marketing Principles. A Case Study

**DOI:** 10.3390/ijerph18073589

**Published:** 2021-03-30

**Authors:** Lee M. Ashton, Megan E. Rollo, Marc Adam, Tracy Burrows, Vanessa A. Shrewsbury, Clare E. Collins

**Affiliations:** 1School of Health Sciences, College of Health, Medicine and Wellbeing, Priority Research Centre for Physical Activity and Nutrition, University of Newcastle, Callaghan, 2308 Newcastle, Australia; megan.rollo@newcastle.edu.au (M.E.R.); tracy.burrows@newcastle.edu.au (T.B.); vanessa.shrewsbury@newcastle.edu.au (V.A.S.); clare.collins@newcastle.edu.au (C.E.C.); 2School of Education, College of Human and Social Futures, Priority Research Centre for Physical Activity and Nutrition, University of Newcastle, Callaghan, 2308 Newcastle, Australia; 3School of Electrical Engineering and Computing, College of Engineering, Science and Environment, University of Newcastle, Callaghan, 2308 Newcastle, Australia; marc.adam@newcastle.edu.au

**Keywords:** google analytics, process evaluation, healthy eating, website, young adults, social marketing.

## Abstract

Background: Reaching and engaging individuals, especially young adults, in web-based prevention programs is challenging. ‘No Money No Time’ (NMNT) is a purpose built, healthy eating website with content and a social marketing strategy designed to reach and engage a young adult (18–34 year olds) target group. The aim of the current study was to conduct a process evaluation of the 12-month social marketing strategy to acquire and engage NMNT users, particularly young adults. Methods: a process evaluation framework for complex interventions was applied to investigate the implementation of the social marketing strategy component, mechanisms of impact and contextual factors. Google Analytics data for the first 12 months of operation (17 July 2019 to 17 July 2020) was evaluated. Results: in year one, 42,413 users from 150+ countries accessed NMNT, with 47.6% aged 18–34 years. The most successful channel for acquiring total users, young adults and return users was via organic search, demonstrating success of our marketing strategies that included a Search Engine Optimisation audit, a content strategy, a backlink strategy and regular promotional activities. For engagement, there was a mean of 4.46 pages viewed per session and mean session duration of 3 min, 35 s. Users clicked a ‘call-to-action’ button to commence the embedded diet quality tool in 25.1% of sessions. The most common device used to access NMNT (63.9%) was smartphone/mobile. Engagement with ‘quick, cheap and healthy recipes’ had the highest page views. Conclusions: findings can inform online nutrition programs, particularly for young adults, and can apply to other digital health programs.

## 1. Introduction

Globally, poor diet quality is the second highest risk factor for mortality in females, resulting in 13.5% [10.8–16.7] of all female deaths in 2019 [1]. For males, it is the third leading risk factor, accounting for 14.6% [12.0–17.6] of all male deaths in 2019 [1]. Optimal diet quality is key to good health, well-being and the prevention of chronic conditions including; cardiovascular disease (CVD), type 2 diabetes, and specific cancers [2,3]. However, improving diet quality represents a major public health challenge at both a national and international level [1]. This is particularly challenging among young adults. A global analysis of adults aged ≥20 years from 187 countries found those aged 20–29 years had the worst diet quality, compared with all other age groups, closely followed by those in the 30–39 year age group [4]. Furthermore, key transitional changes are common during this life stage including changes to employment status (e.g., starting College/University), living situation (e.g., moving away from parents), social environment and influences (e.g., changes to peers and partner relationships), and financial position (e.g., becoming more self-sufficient) which can impact on lifestyle behaviors [5].

Many behavioural interventions have demonstrated efficacy in improving dietary intake but few report ‘real world’ effectiveness data or demonstrate impact over time and across contexts [6,7]. According to one review, it takes an average of 17 years to move 14% of research into clinical practice [8]. As such, a large proportion of health research fails to move beyond pilot studies and small-scale trials to provide ongoing benefits at a population level [9]. 

A collective challenge in preventive health is reaching and engaging individuals, especially young adults who are deemed ‘hard-to-reach’ due to their perceptions that health problems are distal [10]. Failure to recruit sufficient numbers can lead to false negative findings and can impact representativeness of population samples [11]. Online programs have the potential to overcome these barriers, with opportunities to improve reach, accessibility, scalability and achieving cost-effectiveness [12,13]. However, they are commonly reported to have limited ability to engage and retain participants [14]. A review of engagement among 21 online dietary interventions identified low to moderate rates of attrition for those reporting significant positive changes in diet [14]. Therefore, greater understanding of how to reach and engage individuals over sustained periods in a ‘real world’ context is necessary to achieve long-term improvements in diet and have an impact on individual and population health. 

The methods section of this paper describes the development and implementation of a world first purpose built healthy eating website ‘No Money No Time’ (NMNT) http://nmnt.com.au (accessed on 1 February 2021), that factors in time and financial constraints, to improve dietary quality, reduce chronic risk and improve overall health and wellbeing. While the website is freely accessible to all users of the internet, it was developed for a target group—young adults aged 18–34 years. In order to maximise the reach of NMNT and all/target users’ engagement with the website, and to inform the development of future websites with similar aims, in the method section of this paper we describe in detail the process of developing and applying social marketing principles to maximise user reach and engagement with NMNT, particularly users 18–34 years. Social marketing looks to promote public health by applying commercial marketing strategies and has potential to make an impact at the population level [15]. The consensus definition for social marketing is that it: “seeks to develop and integrate marketing concepts with other approaches to influence behaviours that benefit individuals and communities for the greater social good [16].” A helpful guideline to determine the extent of social marketing within a program/intervention has been proposed by Andreasen (2002) [17]. Andreasen’s six benchmark criteria are; behavioural objective, audience segmentation, formative research, exchange, marketing mix and competition. Findings from a recent review of 34 social marketing interventions to improve dietary patterns recommend that application of all six benchmark criteria has greatest potential to improve eating patterns [18].

In complex real world interventions such as NMNT, embedded process evaluation is a key step to identify improvements that can be made before proceeding to impact and outcome evaluation. During the design of NMNT we did not identify any relevant publications that had presented detailed results of the process evaluation of social marketing strategies, implemented to maximise reach and engagement of young adults, in the context of a nutrition website intervention. While the results from the current study have already been used to further improve the NMNT marketing strategies and website, by publishing the methodology, of the approach, results are likely to be useful in guiding other researchers or public health clinicians in conducting similar evaluations.

Therefore, the aim of the current study was to conduct a process evaluation of a 12 month social marketing strategy to acquire and engage NMNT users, particularly the target group of young adults. The evaluation was guided by the UK Medical Research Council process evaluation framework for complex interventions which recommends investigating the implementation components, mechanisms of impact and contextual factors [19].

## 2. Materials and Methods 

### 2.1. Ethical Considerations

Approval for this study was obtained from the University of Newcastle Human Research Ethics committee (H-2018-0512).

### 2.2. Development of No Money No Time (NMNT) 

NMNT (http://nmnt.com.au, accessed on 1 February 2021) is a purpose built website designed to motivate and support individuals to improve overall eating habits (Appendix A). The site was named to reflect the key needs and constraints of end users. The site is freely accessible, but the content and social media marketing strategy specifically targeted to young adults (18–34 years). The site has been informed by formative research with young adults [20,21,22,23,24], our previous intervention research in young adults [25,26] and advice from cross-disciplinary experts in nutrition, human–computer interaction, digital marketing and design. Further, the website content incorporates key behaviour changes techniques (i.e., goal setting, feedback on behaviour, self-monitoring, social comparison) to assist in making positive changes to dietary patterns [6]. A logic model [27,28] illustrating the inputs, activities, outputs, impacts and outcomes planned for the NMNT website is shown in Figure 1. 

The site includes:Personal assessment of diet quality with feedback: the site includes ‘call to actions’ to encourage users to complete the embedded Healthy Eating Quiz (HEQ); a brief, validated tool to assess diet quality [29,30,31,32,33,34]. Once completed, the user obtains a diet quality feedback report that compares individual score with population normative data (Appendix A).Fast, inexpensive recipes for healthy eating: The site also includes over 150 fast, inexpensive and healthy recipes, with one new recipe added each week, that can be filtered based on potential barriers that were identified as part of the scoping work. Filters include ownership of specific kitchen equipment, nutrition motivators, meal type, dietary intolerances, number of serves, cost and dietary preferences (Appendix A).Nutrition ‘hacks, myths and FAQ’ articles: The site has over 100 articles with one new article added each week, that translates nutrition research into lay language to address common dietary misconceptions. Blog topics are informed from common nutrition topics and from a data bank of questions from our Massive Online Open Course (MOOC) “The Science of Weight Loss—Dispelling Diet Myths” from over 57,000 individuals from >180 countries.Personal dashboard: Users are encouraged to sign-up to create a personalised dashboard which enables them to set SMART (Specific, Measureable, Attainable, Realistic and Time—bound) goals relating to their current HEQ diet quality score. The personal dashboard also enables users to store past HEQ results, save favourite recipes and blog articles, and receive recipes based on individual HEQ scores (e.g., if a user scores low in the vegetable sub-group score, then recipes with higher vegetable content will appear in dashboard) (Appendix A).Automated email and social media presence: Those who complete the HEQ and sign-up to an account also receive regular communication via email (approx. 1–2 per month) in order to alert them to new content, recipes and diet improvement hacks on the website. This information is also disseminated on the NMNT social media channels (Twitter, Facebook and Instagram) with approx. 1–2 posts per week.

### 2.3. Development and Implementation Plan for the Social Marketing Strategy to Maximise NMNT User Acquistion and Engagement

A digital marketing consultant alongside the research team devised the social marketing strategy for NMNT. The strategy objective was to maximise the number of new users of NMNT (i.e., user acquisition), particularly within the target age group of 18–34 year olds, and to maximise their interaction with NMNT during the initial visit (i.e., initial engagement) and on an ongoing basis (i.e., longer-term engagement). The strategic approach was to focus on organic traffic, defined as: “visitors coming from search engines, such as Google and does not include paid search advertisements” [35]. The marketing strategy applied all of Andreasen’s social marketing benchmark criteria [17] (Table 1). 

From 17 July 2019 to 17 July 2020 the project manager (L.M.A.) deployed the social marketing strategy with the support of a digital marketing consultant. The following activities were undertaken to enhance the strategy:Implement search Engine Optimisation (SEO) audit: defined as the process of maximising traffic to a website so that it appears high on the search engine results page and is accomplished through use of keywords and best-practice website design [40]. An SEO audit was conducted pre and post launch to increase website visibility on search engines such as Google.Content strategy: regular, high quality content that was optimised for SEO was added to the site (target of one new content piece added each week).Backlink strategy: defined as another website linking to NMNT website [41]. Existing, reputable sites (i.e., Queensland Health) were contacted to enquire about linking to NMNT website.Regular promotional activities conducted: including engaging with media (national and local TV, radio, print) and podcasts to ensure the evidence-based approach to this area of nutrition with links to NMNT website. These activities were facilitated by The University of Newcastle’s communications team preparing a media release in-kind.

### 2.4. Process Evaluation of the NMNT Social Marketing Strategy Over a 12 Month Period

#### 2.4.1. Framework 

NMNT is considered to be a complex intervention due to automated intervention processes based on user’s engagement with the website tools. Based on the UK Medical Research Council guidance for process evaluation of complex interventions [19], this process evaluation investigates: (i) the implementation process for the social marketing strategy); (ii) the mechanisms of impact—user access to and engagement with NMNT considering potential mediators including age group (total sample 18+ years; young adults (18–34 year olds)), user acquisition channel, device and internet browser and (iii) contextual factors.

#### 2.4.2. Data Collection

Google Analytics (Google, California, USA) was installed on NMNT by adding a JavaScript code tracking tag to facilitate data collection related to user recruitment to the website, their characteristics and behavior during website visits from July 2019 to July 2020. Google Analytics collected following user data: 

##### Demographics 

Gender (male, female) and age in years (split into categories: 18–24, 25–34, 35–44, 45–54, 55–64, >65). Age and Gender data is commonly only available for a subset of users as Google Analytics cannot obtain this if the DoubleClick cookie is not available on the users’ browser, or they are not logged into a Google Account, or the Advertising ID is present. Geographical data was collected on all users, based on IP-address. 

##### Engagement Metrics 

Level of engagement was inferred from the following indicators: Bounce rate (%), defined as the percentage of immediate abandonment or single page visits during a session [42]. A low bounce rate indicates that site visitors explore additional content beyond the home page and click deeper into the site and is therefore indicative of high overall engagement [43,44].Mean number of sessions per user. A session is defined as a group of user website interactions that take place within a given period (expires after 30-min of inactivity or at midnight). Usage data (page views, events, social interactions) is all data associated with a session. Mean number of sessions per user is calculated as total sessions divided by total number of users. A higher number of sessions per user is indicative of greater engagement [45].Mean number of page views per session, defined as the number of website pages that the user viewed in a single session. A higher mean number of pages visited indicates higher engagement [45,46]Mean session duration (mins and secs), defined as mean amount of time a user spends on the website in each session. A longer session duration indicates higher engagement [45,46]Goal conversion rate (n and % of session): measures the proportion of sessions that accomplished a pre-defined goal. For NMNT, the goal was set as clicking on a call-to-action button to start the HEQ diet quality survey. Users were required to complete the HEQ to sign-up to an account and access more resources than those who do not. Therefore, starting the HEQ indicated higher engagement.

In addition, to the above listed metrics longer term engagement was defined as the proportion of return users (n and %) and number/duration of website visits from users with the same client ID, determined from cookies which last two years and ‘remembers’ each user [47].

##### Potential Mediators of Engagement 


User acquisition channel: Channels used to access NMNT were categorised into; organic search (e.g., entry to site via search engine such as Google), direct source (e.g., typing the URL of website directly into a browser) referrals via another website, referral via social media and referral via email.Internet browserDevice used to access the website:


##### Contextual Factors

The NMNT website and the social media marketing strategy were designed to appeal to 18–34 year olds. On the 11th March 2020 the World Health Organization declared COVID-19 a pandemic. These factors are considered in interpreting the process evaluation data.

#### 2.4.3. Data Analysis

Google Analytics data were extracted for the total sample and stratified by young adults (18–34 years) and return users. Descriptive statistics were used to analyse user acquisition and user engagement data. In addition, raw data and graphs from Google Analytics is provided in Appendix A. These graphical data were used to observe the impact of the COVID-19 pandemic. 

## 3. Results

### 3.1. Implementation of the Social Marketing Strategy

The social marketing strategy was implemented over 12 months as described in the methods section and this was not affected by the COVID-19 pandemic (as observed in Appendix A which shows consistent patterns of website users throughout the year). Table 2 describes the type and sequence of promotional strategies during this period

### 3.2. Mechanisms of Impact

#### 3.2.1. User Acquisition

In one year, the NMNT website had 42,413 new users from over 150 countries, with the majority from Australia (*n* = 33,841, 79.5%) and female (67.2%) (Table 3). Almost half (47.6%) were in the target age range of 18–34 years. Based on user access data, the most successful promotional strategy was reference to the NMNT site by a CI (CEC) on a national radio program with a young adult target audience. Over the course of the year, three interviews on the radio program generated 2987, 1089 and 1003 users visits on the day of airing (Table 2). 

#### 3.2.2. User Engagement 

##### Total Sample

Over the first year there were 61,273 sessions with an average of 4.46 pages viewed per session (273,170 page views) and an average of 3 min, 35 s per visit. The average bounce rate was 40.0% where users viewed a single page (Appendix A). In a quarter of all sessions (*n* = 15,379, 25.1%) users clicked a call-to-action button to commence the embedded HEQ diet quality assessment tool. In terms of frequency of visits, 93.5% (57,278/61,273) of sessions came from users visiting less than nine times, which suggests disengagement over time (Table 4). Few sessions had a long duration (Table 4), with 65.6% (40,188/61,273) disengaging within the first minute. A funnel report is provided in Appendix A which maps the customer journey and indicates where website users dropped out. The NMNT home page had the highest number of page views (39,606 page views). The content on the home page included brief extracts from key website components to encourage users to engage with the resources and the calls-to-action to motivate users to complete the diet quality assessment tool and log in or sign-up. Subsequently, page 1 of the recipes (total of 14 pages) had the next highest page views 23,121 page views. Engagement with recipes was high with those located on pages one to six all in the top ten pages with the highest number of page views (Table 5). 

##### Return Users 

Around one in five (*n* = 10,067, 19.2%) sessions were return visitors in the 12-month period. There were 24,424 sessions by return users and the number of sessions per return user was higher compared to new users (average of 2.43 vs. 0.87), but number of pages viewed in each session was slightly lower (4.11 vs. 4.69). This was expected, as the return users would have likely viewed some aspects of the site in prior sessions. In addition, the bounce rate was higher in return users (57.7% vs. 28.3%) with goal completions much lower as 6.8% (*n* = 1654) clicked on a call-to-action button to commence the HEQ diet assessment tool compared to 37.3% of new users (Appendix A). Again, this was expected as users were encouraged to complete the dietary assessment at the initial website visit to sign-up to an account. NMNT content viewing was similar to the total sample (Table 5).

##### Young Adults

Age data was available for a subset of users (20,944 sessions or 34% of all total sessions). In these young adults (aged 18–34 years), there were an average of 4.89 pages viewed per session with an average of 3 min, 50 s spent on the site per visit (Appendix A). Mean bounce rate in young adults was 37.5%, while users clicked on the call-to-action button to commence the HEQ in 26.1% of all sessions. The higher engagement and conversion among young adults compared to the rest of sample indicates that targeting to this group was successful (Appendix A). NMNT content viewing was similar to the total sample (Table 5).

##### Potential Mediator—User Acquisition Channel

Total sample: the most common channels for acquiring users included organic search (i.e., entering information into search engine such as Google) (51.9% of total sessions), followed by direct sources (typing URL of website into search bar) (31.6% of total sessions) and referral by websites (10% of total sessions) (Table 6). The most common referral pathways were: i) a public news website: ‘The Conversation’ (23.2% of all referral sessions) and ii) a health insurance company (nib.com.au; the project sponsor) who notified members of the resource (20.3% of all referral sessions). Social media and email channels acquired the lowest proportion of total sessions. The combination of high conversion rate (26.9%), high number of pages viewed per session (mean of 4.6) and long mean session duration (3 min 43 s) among organic searches indicates strong engagement with this approach. Although, referrals from social media sites acquired less traffic at 6.1% (3765/61,273), they had the highest conversion rate (40.7%) and lowest bounce rate (31.1%). Referral via email had highest bounce rate (57.3%), lowest conversion (5.7%) and shortest session duration (mean 2 min 00 s).

Return users (*n* = 10,067): there was a higher proportion of traffic obtained organically (i.e., through search engine) compared to new users (56.7% vs. 48.7%) (Appendix A). However, there was a lower proportion for direct (30.6% vs. 32.2%), referral via website (30.6% vs. 32.2), referral via social media (9.2% vs. 10.5%) and referral via email (0.2% vs. 0.5%). The most common referral pathways for return users were: (i) the public news website ‘The Conversation’ (23.7% of all referral sessions) and (ii) a standalone dietary assessment tool (quiz.healthyeatingquiz.com.au) (21.6% of all referral sessions).

Young adults: observing data from this subset for young adults (aged 18–34 years) compared to all other users (aged 35 years and above) there was a higher proportion of organic traffic (58.7% vs. 49.3%) in young adults (Table 6) but for all other channels there were lower proportions of young adults acquired (Appendix A). The most common referral pathways for young adults aged 18–34 years were: (i) a health insurer (nib.com.au) (16.5% of all referral sessions) and (ii) the standalone brief diet quality assessment tool (quiz.healthyeatingquiz.com.au) (9.6% of all referral sessions). The organic approach appeared to have stronger engagement with young adults compared to the rest of the sample, with higher number of pages viewed per session (mean of 5.22 vs. 5.07), longer mean session duration (4 min 4 s vs. 3 min 56 s) and higher conversion rate (27.2% vs. 26.1%) (Appendix A).

##### Potential Mediators—Internet Browser and Devices

As a proportion of total sessions (*n* = 61,273 sessions), the most popular browsers used were Google Chrome (41.4%) and Safari (40.9%) (Table 7), while smartphone/mobile was the most common device used (63.9%) (Table 8). However, user-experience appears to favour desktop with a lower bounce rate (25%) and higher number of pages per session (6.4) and session duration (5 min, 2 s) in users using this device (*n* = 18,452) compared to mobile and tablet devices (Table 8). In young adults, smartphone/mobile use was higher compared to rest of sample (62.0% vs. 52.7%), while tablet use was much lower (1.9% vs. 10.7%) (Appendix A). In terms of mobile device used, Apple iPhone was most popular for all users and for young adults (Appendix A). 

### 3.3. Contextual Factors

The graphical data presented in Appendix A illustrate that user acquisition was lowest during the summer holiday months in Australia (December to January). These data also indicate that the COVID-19 pandemic, which began to influence the day-to-day lives of Australians from approximately March 2020, did not have a detrimental impact on the response to the social marketing strategy.

## 4. Discussion

The current study provides a detailed process evaluation of a ‘real-world’ healthy eating website (NMNT) over a 12-month period. Overall, the social marketing strategy was deemed to be successful with over 40,000 new users with almost 50% being in the target age group of 18–34 year olds. Our engagement findings are promising. In this section we compare our findings with reference to the most relevant, albeit limited evidence base. 

### 4.1. User Demographics 

NMNT attracted a high proportion of young adults with 11.3% aged 18–24 years and 36.3% aged 25–34 years. This finding confirms the success of the extensive co-design work with young adults and targeted approach for this group to inform the site development. This is especially important given previous challenges in recruiting this ‘hard-to-reach’ group for health programs [48,49]. Despite the lower proportion of males compared to female NMNT users (32.8%), this aligns with findings from a systematic review of 54 dietary interventions among young adults which showed the average proportion of males in gender-neutral programs to be 32% [6]. Traditional masculine ideologies and gender roles may discourage males from seeking help in regard to nutrition (i.e., *real men do not diet*) [50]. Greater efforts are needed in order to recruit young males to healthy eating programs. The high proportion of Apple iPhone users who accessed NMNT provides a rationale for user demographics, as recent research suggests iPhone owners are more likely to be female and younger when compared with Android users [51].

### 4.2. Effectiveness of Marketing Strategies

The social marketing strategy which applied Andreasen’s criteria [17] successfully recruited 42,413 users with a total of 61,273 sessions in 12-months. When compared to published data for other health behaviour platforms broadly, the current findings are similar to those from a website for people with eating disorders (*Break Binge Eating*) with 46,311 users acquired in 13-months [52]. The marketing strategy was similar with a key focus on an organic approach, but there were also paid advertisements implemented. In addition, our user-acquisition is higher compared to other health behaviour websites including one supporting people living with mental illness (*n* = 24,500 users in 12-months) [53] and another focused on improving mental health of young people (*n* = 3076 users in 12-months) [54].

The Google Analytics ‘benchmarking’ feature enables comparison with aggregated industry data from other companies who shared their data during the same time period [55]. Using this tool to compare NMNT with 13,043 websites globally in the industry category ‘food and drink’, it is evident that NMNT had an 8% lower number of new users and 0.06% lower number of sessions compared to average across 13,043 food and drink websites (Appendix A). Despite the marginally lower number of new NMNT users and sessions, this is still considered a success given no paid advertising was used. This is in contrast to the other industry websites which attributed on average 13,986 sessions and 10,239 new users to paid search advertising (Appendix A). As a large proportion of the content included recipes, we limited the Google Benchmarking tool to ‘cooking and recipe’ websites (*n* = 1080 websites). This comparison indicates that NMNT had 16% higher session numbers and 4% higher numbers of new users during this time period (Appendix A).

An organic strategy was a key focus of NMNT. This included an SEO audit pre and post launch, a content strategy (adding one new piece of content per week), a backlink strategy and regular promotional activities implemented throughout the year. The success of this approach was reflected with the highest proportion of total users, young adults and return users being obtained organically. This is particularly important given the established challenges of reaching young adults for health programs [48,49]. A systematic review of 26 weight gain prevention interventions to recruit young adults found many studies utilised traditional approaches (i.e., email and flyers) without use of a social marketing framework [48]. Given there has been a steep rise in eHealth dietary interventions targeting young adults in recent years [6] there is huge potential for utilising these contemporary online marketing strategies. This can be used in combination with ‘traditional/non-digital’ approaches as they may still be an important source of some users. As was observed in NMNT which obtained 5079 users from three national radio interviews on the day of airing. This is consistent with a trial which used integrated information technology with social marketing strategies to recruit young adults to a social and mobile trial [56,57]. Results indicated that combining traditional and contemporary recruitment strategies and using a marketing promotion framework maximised recruitment efforts. 

There was a small proportion of users acquired through social media sources. However, users arriving to the site via social media had the highest conversion rate and lowest bounce rate. This observation was apparent for all users and when stratified by young adults. This finding indicates potential of this approach in future recruitment efforts and especially when recruiting young adults who are considered ‘hard-to-reach’. This is consistent with a scoping review of 30 studies that looked at success of social media to recruit in medical research, and found social media to be the most successful recruitment method for people who are considered ‘hard-to-reach’ and in observational studies [58]. In going forward, greater efforts will be made to utilise this method of recruitment. Specifically, most social media posts on NMNT were static; however, there is potential to test video strategies given that YouTube remains the second largest search-engine [59]. In addition, the social media posts from NMNT were used to direct users to the website where more in-depth information was provided. However, a recent review on social media and nutritional outcomes in young adults suggests this approach may be redundant and other strategies may be required to capitalise on the potential of social media [60]. Specifically, there is a great degree of heterogeneity among young adults with varying levels of engagement (e.g., super sharers, sharers, posters, lurkers and no engagement) [60]. It is suggested that complex interventions need to account for young adults’ variation in social media use and use an array of techniques, such as social support [60], gamification [61] and different messaging [62,63] to enhance engagement. In addition, it is important to consider user feelings around certain foods and how social media content can impact on this. For instance, a recent study implemented a text mining approach to identify feelings expressed with certain foods on twitter and found foods that are negatively perceived are bacon, snacks, red meat, sugar and processed foods, while water, salads, spinach, apples and broccoli were viewed more positively [64]. 

### 4.3. User Engagement 

Engaging users is a common challenge with online programs [14], so several strategies were implemented to enhance engagement including; heightened UX, regular monitoring of site and responding to feedback, new content added each week, multi-channel marketing and encouragement to track progress over time. Engagement data from the current website are promising when compared with published data for other health behaviour platforms. The 273,170 page views from NMNT is higher than platforms that have supported; perinatal women (*n* = 21,619 page views in 12-months) [65], disordered eating (84,054 page views in 13-months) [52], and young people wanting to improve their mental health (*n* = 29,299 page views in 12-months) [54].

Using the Google Analytics benchmarking tool to compare against the average from 13,043 ‘food and drink’ websites [55], it is evident that NMNT had 69% higher average number of pages viewed per session (4.46 vs. 2.63), 119% longer average session duration (3 min, 35 s vs. 1 min 38 s) and a 31.0% lower bounce rate (40.04% vs. 57.99%) (Appendix A). This is encouraging, especially given the high proportion of young adults within our users. When limiting the Google Benchmarking tool to ‘cooking and recipe’ websites (*n* = 1080 websites), NMNT had 113% higher average number of pages viewed per session (4.46 vs. 2.09), 170% longer average session duration (3 min, 35 s vs. 1 min 20 s) and a 39% lower bounce rate (40.04% vs. 65.82%) (Appendix A).

The high proportion of short and single session visits suggest the importance of short, quick-read information, key messages and effective use of call-to-actions to encourage users to covert and carry out the pre-determined goal. This approach was implemented in NMNT, especially for the ‘hacks, myths and FAQs’ section which transforms scientific research into ‘quick-read’ blogs. Readability length was also provided on each post (i.e., “2-min read”) to reinforce this. 

Overall, there was a moderate return rate of 19%, this is lower than the median reported in e-commerce websites which is around 30% [66]. However, it is unclear if this is sufficient for health interventions and especially dietary interventions for young adults as this information is rarely reported in the research literature. A men’s mental health website reported similar return rates prior to advertising (17.7%) [67]. This shows that increased research is needed to improve return rates for health interventions specifically dietary intake in young adults. Possible avenues could include; strategic retargeting ads [67], appropriate balance of email marketing [68], add new content regularly [69] and improve quality and quantity of social media posts [70]. 

### 4.4. Website Improvement 

Exploring user behaviour on the NMNT site has identified areas on the site for improvement. Use of mobile/smartphone was the preferred device to access the website. However, engagement with the site by users accessing via mobile/smartphone appears to be lower than a desktop device. This pattern is comparable to the user-behavior from the 13,043 ‘food and drink’ websites from Google Analytics benchmarking tool (Appendix A). There is a difference in website behavior for mobile device users compared to those that access from a fixed device (e.g., desktop) [71] as users accessing with a mobile device are likely to be doing multiple activities, have a smaller screen size and limited processing power [72]. As such, users ability to “multi-task” on their mobile device may affect user engagement [73]. As the recipe web page on NMNT had the highest page views, it is likely many used the mobile device to access the site while “multi-tasking, possibly at the supermarket, which ultimately affected engagement. Overall, it is imperative to prioritise formatting to heighten user-experience on mobile devices during website development. The high engagement with the recipe section on NMNT affirms that users are looking for ideas on cheap, quick and healthy recipes. Going forward, this can be used in future recruitment efforts but also suggests more focus on improving other areas of the website.

### 4.5. Strengths and Limitations

Strengths of the current study includes; a comprehensive data set of users that is comparable to aggregated industry data from ‘food and drink’ websites. In particular, there was a high number of young adults who engaged with the website. There is opportunity to use data from this study to further engage this ‘hard-to-reach’ group. Another strength is comprehensive use of a social marketing strategy that adhered to social marketing benchmark criteria. Furthermore, use of Google Analytics to explore quantitative website usage data can be leveraged for continual website improvement. Use of the MRC process evaluation framework for complex interventions was helpful and highlights the importance of investigating the influence of contextual factors, e.g., COVD-19 pandemic on our social marketing Despite this, there are some limitations to acknowledge. Firstly, due to the nature of the study design there is potential for sampling bias which limits the generalizability of findings. In addition, stratification by age was only available for a subset of users as Google Analytics as age can only be determined if a user is logged into their Google account, or if the DoubleClick cookie it not switched off. Therefore, results from young adults may not fully represent the overall composition of traffic. In addition, the number of users may be inaccurate as a new client ID is created each time a user operates a different browser, deletes browser cookies or switches devices. This results in the same user being counted as a new user [45]. Furthermore, there are limitations to ascertaining engagement indicators (e.g., pages per session and session duration). Many pages viewed per session may be the result of superficial exploration of website pages, while a long session duration could be due to webpages being left open while engaging in other irrelevant activities [54]. Google Analytics is the most common third party service for collecting information on user behaviour in online health programs [74] and was used in the current study to measure reach and engagement due to ease of tracking and extracting data. However, capturing usage data more specific to the platform will require additional visualisation tools [75] and engagement indices such as eye-tracking techniques or surveys to get better insights into the data [76,77,78,79].

## 5. Recommendations 

Recommendations for future improvement of NMNT platform include targeting efforts to improve the return rate through retargeting ads, balanced email marketing, more frequent content addition and improved quality and quantity of social media posts. Furthermore, improving user-experience on mobile devices will be prioritised. Additional recommendations to inform future online heathy eating programs for young adults and potentially applicable to other digital health programs, include;
Utilising social marketing frameworks (e.g., Andreasen’s six benchmark criteria [17]) to guide an overall marketing strategy.Seek advice from cross-disciplinary experts (e.g., human–computer interaction, digital marketing and design)Undertake extensive co-design with the target audienceDrive organic traffic to the site as a sustainable way of obtaining users/participants long term.Continually monitor and evaluate website metrics using Google Analytics to identify issues with reach or engagement and act accordingly.If using social media related to the website, consider young adults’ social media use patterns and use relevant techniques to enhance engagement (e.g., different types of messaging or gamification).

## 6. Conclusions

The current process evaluation of a ‘real-world’ healthy eating website (NMNT) provided important information about marketing strategies, user-engagement and aspects of the platform for future improvement. In terms of user-acquisition, an organic approach was successful in reaching total users, young adults and return users. This approach included an SEO audit pre and post launch, a content strategy, a backlink strategy and regular promotional activities implemented throughout the year. There was a high level of engagement by users with NMNT compared to aggregated industry data from similar websites. Several strategies were implemented to improve user-engagement, including enhanced UX, regular site monitoring and responding to feedback, new content added weekly, multi-channel marketing and encouragement to track progress over time. Findings can inform online nutrition programs, particularly for young adults, and can apply to other digital health programs.

## Figures and Tables

**Figure 1 ijerph-18-03589-f001:**
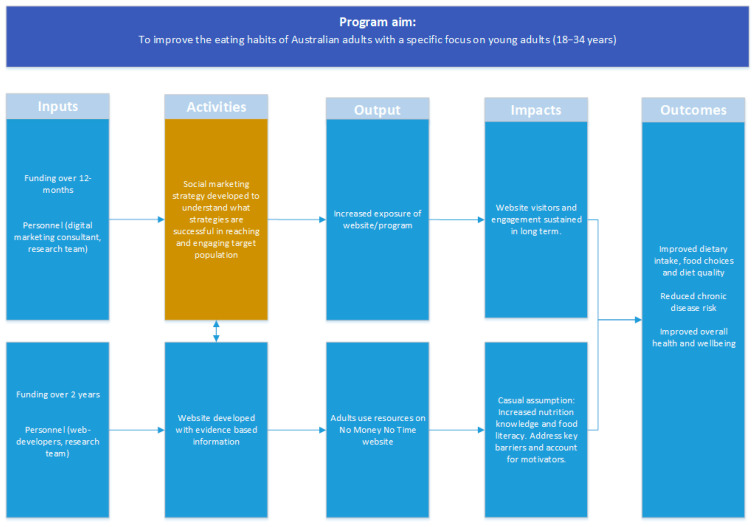
Logic model for the NMNT illustrating the role of the social marketing strategy.

**Table 1 ijerph-18-03589-t001:** Application of Andreasen’s social marketing benchmark criteria [17] in No Money No Time.

Andreasen’s Social Marketing Benchmark Criteria	Brief Description of Benchmark Criteria	How We Addressed the Criteria in No Money No Time
Behavioural objective	The objective of social marketing is to change people’s behavior.	Goal: To help people to improve diet quality measured using HEQ score.
Audience segmentation	Segmentation investigates key needs and motives for unique groups to inform different marketing and promotion mixes accordingly.	Identification of unique groups based on key motivators, individually targeted using personalised emails, recipes and blog articles specific to motivators. These were: (i) to achieve or maintain a healthy weight, (ii) to find out whether my diet is healthy (iii) to perform better in sport (iv) to know more about how to eat better and (v) to feel better or improve well-being.
Formative research	Formative research investigates the consumers’ needs and provide understanding of motives that can be influenced to achieve desired behavior change goals.	Previous formative research on needs, motivators, preferences and barriers undertaken with target audience (young adults) [20,21,22,23,24].
Exchange	Social marketing exchange reduces effort for users and emphasises/maximises the consumer benefit, e.g., “what would motivate people to engage willingly with the website and offer them something beneficial in return” [36]	Goal—increase user engagement through: Heightened user experience (UX): i.e., the process of supporting user behaviour through usability, usefulness, and desirability provided in the interaction with a website [37]. A pre-launch site audit of user-experience (UX) was conducted. Young adults (*n* = 30) tested the site prior to launch to further optimise UX for the needs of intended audience. Continual site monitoring and response to feedback: obtained via an embedded evaluation tool (HotJar) to help understand users website behaviour and to obtain feedback through feedback polls, session recordings and heatmaps to visualise popular (hot) and unpopular (cold) webpage elements [38]. This assisted in prioritising enhancements for ongoing improvements to UX and user-engagement, with enhancements deployed sequentially. New weekly content to address trending diet topics. Multi-channel marketing, defined as the implementation of a single strategy across multiple channels for better engagement with users/participants [39]. These included: automated emails (approx. 1–2 emails per month to those who complete the HEQ and sign-up to account) and regular (1–2 posts per week) social media presence (Twitter, Instagram and Facebook) to inform of new content and encourage users to return to the site. Encouragement to track progress over time by email and website messaging to re-take the HEQ every 3 months and track progress over time.
Marketing mix	Moving beyond communication using multifaceted interventions (e.g., more than promotion and communication). Refers to many benefits for the target audience considering the 4p’s (product, place, price and promotion) [36]	Benefits for user includes: institutional trust, free access to website, evidence based information from accredited practicing dietitians and nutrition researchers, personalised support (i.e., dietary feedback report which compares score to Australian normative data). Benefits promoted on site and in marketing materials. Conversion Rate Optimization (CRO) tested UX variations and adjusted marketing strategies to increase percentage of website visitors, sign up rate and HEQ completions.
Competition	Competition faced by the desired behavior. Competition could be harmful behaviors that will lead to this behavior or any behavior, product or idea that negatively impacts health and wellbeing [36]	Analysis of direct and indirect competition.

**Table 2 ijerph-18-03589-t002:** Promotional strategy by number of users visiting site.

Scheme 2	Publication/Release Date	Number of Users on the Day of Release/Pub
Local Radio interview, local Newspaper article x2, online blog x2	18/07/2019	271
Local Newspaper article	19/07/2019	431
Email campaign	12/08/2019	35
Local Newspaper article	22/08/2019	47
Academic news article	27/08/2019	105
Link to site on Massive Online Open Course	4/09/2019	22
Podcast	24/09/2019	618
Podcast shared on social media by podcast creator	25/09/2019	512
National radio interview (number 1)	17/10/2019	1089
Local Newspaper article and local TV news segment	11/02/2020	664
National radio interview (number 2)	20/02/2020	1003
Online article in International news resource	20/03/2020	112
Promoted by state health organisation on social media	24/03/2020	544
Local newspaper article x2	4/04/2020	120
National radio interview (number 3)	9/07/2020	2987

**Table 3 ijerph-18-03589-t003:** Demographics of new users.

Demographics	Category	Total, *n* (% of Users)
Country (150 countries)	Australia	33,841 (79.5%)
	United Kingdom	2795 (6.6%)
	United States	1582 (3.7%)
	South Africa	685 (1.6%)
	New Zealand	536 (1.3%)
	Other ^≠^ (145 other countries)	2974 (7.3%)
Age range (*n* = 15,258 *)	18–24	1738 (11.3%)
	25–34	5605 (36.3%)
	35–44	2964 (19.2%)
	45–54	2020 (13.1%)
	55–64	1682 (10.9%)
	65+	1425 (9.2%)
Gender (*n* = 15,574 *)	Female	10,530 (67.2%)
	Male	5133 (32.8%)

* Age and Gender data is only available for a subset of users (see methods section for further detail). ^≠^ Individual countries accounting for less than 1% of users.

**Table 4 ijerph-18-03589-t004:** Frequency and duration of visits.

Session	Full Sample (*n* = 61,273), *n*%	Young Adults (*n* = 10,418), *n*%
1	36,849 (60.1%)	6232 (59.8%)
2	9620 (15.7%)	1646 (15.8%)
3	4276 (7.0%)	747 (7.2%)
4	2453 (4.0%)	431 (4.1%)
5	1588 (2.6%)	275 (2.6%)
6	1097 (1.8%)	188 (1.8%)
7	796 (1.3%)	119 (1.1%)
8	599 (1.0%)	92 (0.9%)
9–14	1878 (3.1%)	326 (3.1%)
15–25	1017 (1.7%)	188 (1.8%)
26–50	524 (0.9%)	66 (0.6%)
51–100	318 (0.5%)	53 (0.5%)
101–200	247 (0.4%)	48 (0.5%)
201+	11 (0.0%)	0 (0%)
Session Duration (mins)		
≤1 min	40,188 (65.6%)	6665 (64.0%)
>1 min to ≤3 min	6880 (11.2%)	1230 (11.8%)
>3 min to ≤10 min	6949 (11.3%)	1220 (11.7%)
>10 min	7256 (11.8%)	1303 (12.5%)

**Table 5 ijerph-18-03589-t005:** Top ten page views for total sample, young adults and for return users.

Total Sample (18+ years)		Young Adults (18–34 year olds)		Return Users	
Page	Total (% of Total Page Views: 273,170)	Page	Total (% of Total Page Views: 39,363)	Page	Total (% of Total Page Views: 100,317)
Homepage	39,606 (14.5%)	Homepage	7557 (19.2%)	Homepage	10,679 (10.7%)
Recipes page 1	23,121 (8.5%)	Recipes page 1	4303 (10.9%)	Recipes page 1	8729 (8.7%)
Recipes page 2	6776 (2.5%)	Recipes page 2	1185 (3.0%)	Recipes page 2	2254 (2.3%)
Individual recipe	5191 (1.9%)	Individual recipe	970 (2.5%)	Recipes page 3	1871 (1.9%)
Recipes page 3	5181 (1.9%)	Hacks, Myths, FAQ’s	938 (2.4%)	Hacks, Myths, FAQ’s	1858 (1.9%)
Hacks, Myths, FAQ’s	4953 (1.8%)	Recipes page 3	886 (2.3%)	Individual recipe	1537 (1.5%)
Recipes page 4	3907 (1.4%)	Recipes page 4	648 (1.7%)	Recipes page 4	1468 (1.5%)
Everyday superfoods	3362 (1.2%)	Everyday superfoods	571 (1.5%)	Everyday superfoods	1296 (1.3%)
Recipes page 5	3048 (1.1%)	Recipes page 5	524 (1.3%)	Recipes page 5	1159 (1.2%)
Recipes page 6	2444 (0.9%)	Recipes made with microwave	487 (1.2%)	Recipes page 6	931 (0.9%)

**Table 6 ijerph-18-03589-t006:** User acquisition channels and engagement metrics.

	Total Sample (18+ years)					Young Adults (18–34 years)				
Channels	Sessions (*n* = 61,273), *n* (%)	Bounce Rate	Pages per Session, *n*	Mean Session Duration (min:s)	Conversions, *n* Rate (%)	Sessions (*n* = 10,418), *n* (%)	Bounce Rate	Pages per Session, *n*	Mean Session Duration (min:s)	Conversion Rate (%)
Organic Search	31,770 (51.9%)	41.8%	4.6	3:43	26.9%	6117 (58.7%)	38.7	5.2	4:04	27.2%
Direct	19,360 (31.6%)	39.4%	4.5	3:35	20.7%	2318 (22.3%)	36.3	4.6	3:38	24.2%
Referral	6115 (10.0%)	37.7%	4.6	3:47	20.7%	1212 (11.6%)	37.8%	4.7	3:46	18.7%
Social media	3765 (6.1%)	31.1%	3.0	2:12	40.7%	758 (7.3%)	31.7%	3.4	2:44	36.8%
Email	262 (0.4%)	57.3%	2.4	2:00	5.7%	13 (0.1%)	46.2%	2.6	0:23	7.7%

**Table 7 ijerph-18-03589-t007:** Proportion of total sessions by internet browser used to access No Money No Time.

Total Sample (18+ years)		Young Adults (18–34 year olds)	
Browser	*n* (% of Total Sessions: 61,273)	Browser	*n* (% of Sessions for this Age Group—10,418)
Google Chrome	25,391 (41.4%)	Google Chrome	7841 (75.3%)
Safari	25,083 (40.9%)	Samsung Internet	823 (7.9%)
Samsung internet	2883 (4.7%)	Safari (in-app)	528 (5.1%)
Safari (in-app)	2329 (3.8%)	Android Webview	401 (3.8%)
Edge	1455 (2.4%)	Safari	364 (3.5%)
Other (15 other browsers)	4132 (6.8%)	Other (4 other browsers)	461 (4.4%)

**Table 8 ijerph-18-03589-t008:** Proportion of total sessions by device.

	Total Sample (18+ years)					Young Adults (18–34 year olds)				
Device	Sessions (*n* = 61,273), *n* (%)	Bounce Rate	Pages per Session, *n*	Mean Session Duration	Conversions, *n* Rate (%)	Sessions (*n* = 10,418), *n* (%)	Bounce Rate	Pages per Session, *n*	Mean Session Duration	Conversion Rate (%)
Mobile (phone)	39,137 (63.9%)	47.5%	3.5	2 min, 50 s	24.0%	6457 (62.0%)	46.2%	3.6	2 min, 49 s	24.9%
Desktop (inc laptop)	18,452 (30.1%)	25.0%	6.4	5 min, 2 s	28.0%	3762 (36.1%)	22.3%	7.0	5 min 33 s	28.7%
Tablet	3684 (6.0%)	36.5%	5.3	4 min, 18 s	21.9%	199 (1.9%)	43.7%	5.5	4 min, 06 s	21.1%

## Data Availability

Data available on request due to restrictions. The data presented in this study are available on request from the corresponding author. The data are not publicly available due to ethical reasons.

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
