# Peer review of "Process Evaluation of the ‘No Money No Time’ Healthy Eating Website Promoted Using Social Marketing Principles. A Case Study"

_ijerph, 2021, doi:10.3390/ijerph18073589_

Round 1
Reviewer 1 Report
As this is the second review of the paper (after updated done by the Authors) I put some general comments and refer to the parts changed by the Authors. I find the current version of the paper significantly improved.
Aim of the paper
What is the most important – aim of the study is much more clear and, in my opinion, was reached by the Authors. Additionally, the Authors defined the research gap.
Methods and data analysis
With a logic model of NMNT methodological section provides more value to the reader. Methodological part and data analysis were significantly updated (e.g. 3.1.; 3.2.)
Strengths and limitations
I would consider putting “Strengths and limitations” as a last part of the paper, after conclusions.
Conclusions
As you have extended the discussion part, “Conclusions” section seems to be quite general now. I would consider providing more recommendations to the health promotion programs and digital social campaigns (now recommendations are present in only the last sentence of the “conclusions”). You can also consider changing the name of the section to “Conclusions and recommendations”.
Author Response
We thank the reviewer for their constructive critique of the manuscript. We believe the suggested changes have greatly improved the quality of the manuscript.
Please see the attachment for point-by-point response to reviewers comments
Many thanks

Reviewer 2 Report
I thank the authors for this revision work, which satisfies most of my concerns about the paper.
Author Response
We thank the reviewer for reviewing our manuscript. No changes were required.
Many thanks
Reviewer 3 Report
Please, read the attached document

Author Response
We thank the reviewer for their constructive critique of the manuscript. We believe the suggested changes have greatly improved the quality of the manuscript.
Please see the attachment for point-by-point response to reviewers comments
Many thanks

This manuscript is a resubmission of an earlier submission. The following is a list of the peer review reports and author responses from that submission.
Round 1
Reviewer 1 Report
The presented paper deals with an important topic. As the Authors highlighted in the Introduction “Globally, poor diet quality is the second-highest risk factor attributable for deaths in females, resulting in 3·48 million deaths, or 13·5% of all female 31 deaths in 2019”. Then, the Authors stated (what is supported by other studies) that behavioral interventions demonstrated high efficacy in improving dietary intake. Finally, as it is suggested, “online programs” and “social marketing” seems to be sensible tools for (previously mentioned) behavioral interventions.
Additionally, I believe the set of data collected by the Authors allows for complex analysis and important findings.
The strengths of the paper are as follows:
- number of cases in a database
- time scope of the study
However, in my opinion, the whole paper needs to be carefully reconsidered and restructure. I will present all my doubts/recommendations in the following sections:
Section 1: the aim of the study
The aim declaration is following:
Therefore, the aim of the current study was to conduct a detailed process evaluation of NMNT by exploring the attributes related to successful reach and user-engagement and stratified for young adults (18-34 years) over a 12-month period. Specific objectives were to summarise demographics of NMNT users; evaluate the most successful social marketing strategies for acquiring users; evaluate level of user-engagement with the NMNT platform and identify areas for improvement for the NMNT platform.
The current aim is rather unclear, it is difficult to understand what the main goal of the study is. Based on the first part of the statement “to conduct a detailed process evaluation of NMNT” the goal seems to be the NMNT evaluation. But, in the next words “by exploring the attributes related to successful reach and user-engagement” attention is paid to the identification of the success-factors of “reach and user-engagement”. Those two topics are very different and, from a methodological point of view, should be organized differently.
The Authors tried to be more specific in the “Specific objectives” – however it makes the aim declaration even more difficult.
What is needed in the paper is to revise the aim and clearly state the problem of the paper and the research gap (which is not presented now). Based on the text and presented data, I believe the best option will be to focus on the most efficient tools for building user engagement and reach. As I understand Authors should have knowledge about promotional activities used to support NMNT. If so, it would be possible to present, e.g. costs of particular promotional tools and number of new users; additional time on the website, etc.
The efficiency approach instead effectiveness approach would provide much more insightful results.
Section 2: method
Lack of a clear statement of the study aim results in the ambiguous method. In the current version of the paper the whole “2. Materials and Methods” paragraph contains in fact:
- description of the NMNT, i.e. 2.2.; 2.3.; 2.3.2.;
- information about how the data were collected, i.e. 2.3.1.; 2.4. (the whole section); 2.5.
Some sub-paragraphs have a misleading title, e.g. “2.3.1. Recruitment” contains, data collection approach and rather promotional (not, it is stated in the text, marketing) strategies.
What important sub-paragraph “2.4.2. Recruitment and effectiveness of marketing strategies (Google Analytics)” suggest it is related to the recruitment process (do we have two “recruitments”?). The word “effectiveness” is inappropriate, as 2.4.2. presents just how the NMNT was reached. Effectiveness is a relation of input to the outcome
Section 3: Results and discussion
Now the presentation of results is a kind of description of an NMNT based on different sources. From time to time Authors jump to the younger group of users.
Paragraph “4. Discussion” is in my opinion rather results presentation than a typical discussion as the Authors compare mainly some Google Analytics indicators for NMNT and other websites “in the industry category food and drink”.
Section 4: More specific comments
Lines 44-45 => might be used for research gap declaration.
Author Response
We thank the reviewer for their constructive critique of the manuscript. We believe the suggested changes have greatly improved the quality of the manuscript. Please find response to each comment in attached.
Many thanks,
Dr Lee Ashton

Reviewer 2 Report
The article in its current version reports the results from a website and the efforts to increase reach and engagement. To be published, it is suggested to incorporate more robust comparison metrics in the text (eg, using data from some similar websites) and to cite more marketing and websites literature to justify the metrics used, the selected target market, gender and age differences, Apple users vs. other users behaviors, etc. Additionally: - Justify the selected target market (

Author Response

(The authors gave the same response as above.)

Reviewer 3 Report
please see attached.

Author Response

(The authors gave the same response as above.)
